# Waste to Carbon: Influence of Structural Modification on VOC Emission Kinetics from Stored Carbonized Refuse-Derived Fuel

**Andrzej Białowiec [1],\*** , **Monika Micuda [1]** , **Antoni Szumny [2]** , **Jacek Łyczko [2]**
and **Jacek A. Koziel [3]**

[1] Faculty of Life Sciences and Technology, Wrocław University of Environmental and Life Sciences,
   50-375 Wrocław, Poland; micuda.monika@gmail.com
[2] Faculty of Biotechnology and Food Science, Wrocław University of Environmental and Life Sciences,
   50-375 Wrocław, Poland; antoni.szumny@upwr.edu.pl (A.S.); jacek.lyczko@upwr.edu.pl (J.Ł.)
[3] Department of Agricultural and Biosystems Engineering, Iowa State University, Ames, IA 50011, USA;
   koziel@iastate.edu
\* Correspondence: andrzej.bialowiec@upwr.edu.pl; Tel.: +48-71-320-5973

**Abstract:** The torrefaction of municipal solid waste is one of the solutions related to the Waste to Carbon concept, where high-quality fuel—carbonized refuse-derived fuel (CRDF)—is produced. An identified potential problem is the emission of volatile organic compounds (VOCs) during CRDF storage. Kinetic emission parameters have not yet been determined. It was also shown that CRDF can be pelletized for energy densification and reduced volume during storage and transportation. Thus, our working hypothesis was that structural modification (via pelletization) might mitigate VOC emissions and influence emission kinetics during CRDF storage. Two scenarios of CRDF structural modification on VOC emission kinetics were tested, (i) pelletization and (ii) pelletization with 10% binder addition and compared to ground (loose) CRDF (control). VOC emissions from simulated sealed CRDF storage were measured with headspace solid-phase microextraction and gas chromatography–mass spectrometry. It was found that total VOC emissions from stored CRDF follow the first-order kinetic model for both ground and pelletized material, while individual VOC emissions may deviate from this model. Pelletization significantly decreased (63%~86%) the maximum total VOC emission potential from stored CDRF. Research on improved sustainable CRDF storage is warranted. This could involve VOC emission mechanisms and environmental-risk management.

**Keywords:** waste management; municipal solid waste; renewable fuel; torrefaction; volatile organic compounds; emissions control; occupational safety; densification; resource recovery; circular economy

## 1. Introduction

### 1.1. Waste-Management Policies

Waste management is a complex system of strategies, technologies, and procedures that were designed by experts, aiming to solve local, regional, and global problems under guiding socioeconomic and environmental principles. In general, two main approaches exist in waste management. The first is focused on economic opportunity. Waste management can be a source of income because customers have to pay for this service, and waste is a resource of valuable recoverable materials. It is a mass-scale business that is prospective due to demographic and economic development. The second approach focuses on environmental protection and, above all, human well-being. Waste poses a threat to

the environment and human life, and there is a need to live in a clean and healthy environment. Both approaches rely on scientific principles that were considered by the European Union (EU) to develop waste-management policies. In the short term, i.e., by 2020, the EU has two main objectives [1]: to reduce the mass of biodegradable landfilled waste by 65%, and to reach 50% by weight recycling of packages.

EU waste-management policies implemented so far resulted in a decrease in landfilling share and an increase in recycling share in mass balance. Thermal-waste treatment is also slightly growing, mainly due to investments in recent years in new member states, including Poland [2]. In 2015, the Circular Economy Package [3], a policy related to the circular economy, was implemented, aiming to achieve the following goals by 2030: 65% recycling of all municipal waste; 75% recycling of packages; limiting municipal-waste landfilling to 10% by weight; and eliminating package landfilling. By 2050, full implementation of a circular economy, and the zero-waste concept, which assumes that nothing is 'wasted', should be achieved.

The implementation of such ambitious policies results in a significant phase-out of technologies that are redundant, e.g., the collection of mixed waste, mechanical–biological waste treatment, and waste incineration or landfilling [4]. On the other hand, technologies related to selective collection, recycling, and the concepts of 'Waste to Energy' [5–8] or 'Waste to Carbon' [9] would be promoted.

## 1.2. Waste to Carbon Concept in Waste Management

The Waste to Carbon concept is the conversion of organic waste into valuable materials, including fuel with high carbon concentration. Low-temperature thermal-treatment technologies, such as torrefaction, were adapted for organic waste [10–12]. To date, torrefaction was mainly used for the improvement of biomass properties [13]. One of the proposed pathways for the waste-to-carbon process is the torrefaction of municipal solid waste (MSW), which turns it into carbonized refuse-derived fuel (CRDF) with a calorific value between 24 and 26 MJ·kg$^{-1}$ [14]. In our previous study, significant increase of lower heating value (LHV) during torrefaction was found. Białowiec et al. [14] showed an increase of LHV from 19.6 MJ·kg$^{-1}$ in raw RDF to 25.3 MJ·kg$^{-1}$ in CRDF. Białowiec et al. [9] also showed the feasibility of CRDF pelletization. In this case, the lower calorific value increased from 21.0 [15] to 25.9 MJ·kg$^{-1}$. Edo et al. [16] also indicated an increase of Refuse-Derived Fuel (RDF) LHV from 19.7 to 21.2 MJ·kg$^{-1}$ due to 90 min torrefaction at 220 °C. The evidence is growing that CRDF can be a replacement for bitumen and lignite coal or woody biomass [14]. CRDF solves the problem of MSW management, and particularly MSW of organic origin. Waste can be transformed into CRDF, thus not being landfilled and making it useable as fuel.

## 1.3. Volatile Organic Compound (VOC) Emissions from Biochar

One of the remaining issues related to the development of torrefaction technology for MSW is mitigating the potential impact of VOC emissions from biochar, including CRDF [17]. Many VOCs cause negative effects on human life and health. According to EU law, a VOC is defined as any organic compound or creosote fraction that, at 293.15 K, has a vapor pressure of no less than 0.01 kPa, or having equivalent volatility under specific conditions of use [18]. However, in EU legislation there is also another definition for VOC, that is, any organic compound with an initial boiling point of less than or equal to 250 °C, measured at a standard pressure of 101.3 kPa [19]. In turn, in the United States, VOCs are defined as any carbon compound, excluding CO, $CO_2$, $H_2CO_3$, metal carbides or carbonates, and $(NH_4)_2CO_3$, which participates in atmospheric photochemical reactions [20]. The most common VOCs are halogenated compounds, aldehydes, alcohols, ketones, aromatic compounds, and ethers [21]. The World Health Organization [22] classifies organic pollutants into three groups by boiling point: very VOCs (VVOCs), 0 to 50~100 °C; VOCs, 50~100 to 240~260 °C; and semi-VOCs (SVOCs), 240~260 to 380~400 °C.

The health effects of VOC emissions from CRDF could be concerning, especially when considering inhalation during production, handling, storage, transportation, and use. Some VOCs are classified

as highly reactive, mutagenic, and carcinogenic. Even small concentrations of these VOCs can cause health problems and chronic diseases, and even be fatal. According to a study conducted by the U.S. Environmental Protection Agency [23], VOCs are responsible for 35%~55% of lung-cancer risk [24] and contribute to photochemical smog [25]. It was proven that VOCs can directly inhibit microbiological and plant processes [26,27]. A provision was made that VOC emissions should be halved from the released amount in 2000 [25] as a consequence of increased knowledge and awareness of their negative impact on human health and the environment in the Goteborg Protocol.

To date, there are only a few studies on the qualitative and quantitative analysis of VOCs in biochars derived from biomass. The occurrence of up to 140 VOCs was reported [26], of which 74 were identified. The most frequently observed compounds in biochar are acetone, benzene, methyl ethyl ketone, toluene, methyl acetate, ethanol, phenol, and cresols. According to Buss et al. [28], biochars are associated with aliphatic acids and naphthalene. Char Team 2015 reported 26 VOCs [29]. The potential problem of VOCs in biochar was also noticed by Taherymoosavi et al. [30], who analyzed biochar from composted MSW. Compost was thermally processed at temperatures ranging from 450 to 650 °C. Different VOC types were detected, including alkylbenzenes, methoxyalkylphenols, N-containing VOCs, furans, and the BTEX group. Their percentage share varied depending on thermal-process conditions. The generation of the BTEX group to obtain biocarbon from raw materials is of significant concern. The highest concentrations were found for naphthalene, toluene, phenol, benzonitrile, and several compounds with the ester group. Wang et al. [31] reported the presence of polycyclic aromatic hydrocarbons (PAHs), with concentration ranging from <0.1 to >10,000 mg·kg$^{-1}$. The highest concentrations were found for naphthalene and phenanthrene.

In recent work, Białowiec et al. [17] found 84 VOCs in the headspace of CRDF produced from municipal waste, including many that are classified as toxic, e.g., benzene or toluene derivatives. The highest emission was measured for nonanal, octanal, heptanal. The top 10 most emitted VOCs consisted of almost 65% of total emissions. The mixture of emissions from CRDF VOCs differed from that emitted by other types of biochars, produced from different types of feedstock, and under different pyrolysis/torrefaction conditions [26,28].

### 1.4. Structural Modification of Biochar as VOC Emission Mitigation Method

In this paper, structural modification via densification (e.g., pelletization) is proposed as a mitigation strategy for VOC emissions from CRDF. Mitigation of VOC emissions from CRDF (postproduction) could be also addressed by other approaches. such as pretreatment of feedstock, treatment of char with chemicals, thermal treatment, or microbiological processing, all of which could be explored in research. The prospect of CRDF utilization suffers from having low bulk density and would therefore incur higher transportation and storage costs. The pelletizing process increases mechanical strength and lowers bulk density [9] while increasing energy density. Compaction of biomass into briquettes and pellets is an old process that has been known for more than 130 years. The benefits of biomass densification were widely presented and discussed in recent review papers [32–34], including the integration of torrefaction with pelletization [35]. Pelletizing technology is mature from production to end use, so pelletized CRDF can help by adopting it wide-scale in waste management. Pelletizing increases material grindability [36]. Pelletization has been used to improve fuel properties as, in addition to increased energy density, humidity decreases, and regular shapes facilitate transport and subsequent burning in boilers. Additives are often used during pelletizing for improved compaction and material binding. Binders improve pellet durability and physical quality, reduce dust potential, improve pelletizing efficiency, and reduce energy costs [37]. A feasible additive is sodium silica, known as 'water glass' (WG), which has been used for preparation briquettes from coal [38]. Thus, analogous to coal, for CRDF pelletizing, the use of sodium silica as a binder and coating was proposed by Białowiec et al. [9]. It was determined that the optimum pressure for CRDF pelletization is 50.8 MPa, and that 10% addition of water glass does not improve CRDF pellet durability.

*1.5. Objectives*

This research aimed to address the gap in knowledge to improve the environmental safety of CRDF storage. Specifically, the key question it aims to answer is how CRDF structural modification (i.e., pelletization with and without a binder) affects VOC emission kinetics during storage. The kinetics of VOC emissions from stored CRDF has never before been determined. Additionally, for the first time, the effect of CRDF densification on the VOC emissions was investigated. Results from this research are needed to develop improved strategies for CRDF scaling up and adoption as a future-proof technology for resource recovery that is consistent with zero-waste and circular-economy goals.

## 2. Materials and Methods

*2.1. CRDF Production*

Details of CRDF production and its properties were described in detail elsewhere [9]. Briefly, a flammable MSW fraction was torrefied to CRDF and then subjected to structural modification, including grinding [16], and pelletization with 50.8 MPa optimized pressure [9] with and without binder addition. The resulting CRDF pellets were 12 mm diameter x 48 mm length and weighed 2 g.

*2.2. Qualitative and Quantitative Analyses of VOC Emitted from Stored CRDF*

VOC measurements were made using headspace (HS) solid-phase microextraction (SPME) for gas extraction, and gas chromatography coupled with mass spectrometry (GC–MS) for analysis; this was described in detail elsewhere [17]. SPME is a 'green', solventless technology that combines VOC sampling and sample preparation. Briefly, 2-undecanone was used as an internal standard, and a universal SPME fiber 3-component DVB/CAR/PDMS 50/30 μm coating was used for 20 min of VOC sampling from the sealed headspace of CRDF stored in 1000 mL glass jars at 23 °C after Day 1, 2, 3, 4, and 7. Separation, identification, and quantification were completed with GC–MS described by Bialowiec et al. [17].

*2.3. Estimating Kinetic Parameters of VOC Emissions from Stored CRDF*

The volatile organic compound emissions were estimated using a first-order cumulative model:

$$E = E_0 \cdot \left(1 - e^{(-k \cdot t)}\right), \tag{1}$$

where $E$ = VOC emission after storage time $t$, $\mu g \cdot kg^{-1}$ (VOC·CRDF$^{-1}$); $k$ = emission-rate constant, day$^{-1}$; $E_0$ = maximum emission potential, $\mu g \cdot kg^{-1}$ (VOC·CRDF$^{-1}$); and $t$ = time, day. Standard error for $E_0$ and $k$ was also estimated.

Additionally, emission half-time ($t_{0.5}$; day) was estimated with

$$t_{0.5} = \frac{ln2}{k}. \tag{2}$$

The emission rate ($r$, $\mu g \cdot (kg \cdot day)^{-1}$) was estimated with

$$r = E_0 \cdot k. \tag{3}$$

The nonlinear regression estimation of VOC emissions was completed with the application of Statistica 12 software (StatSoft, Inc., TIBCO Software Inc., Palo Alto, CA, USA). The 75% threshold to fit estimated parameters to the experimental data was used as significant. This means that the first-order kinetics of a VOC emission were applied when $R^2$ value was >0.75. Similar criteria were used for emission-correlation analyses with the physicochemical properties for each VOC. The working hypothesis was that the emission-rate constant would increase with the VOC boiling point.

## 3. Results

The volatile organic compound emissions from stored CRDF followed first-order kinetics for all tested variants (i.e., loose (control) and pelletized CRDF (treatment) (Figure 1)).

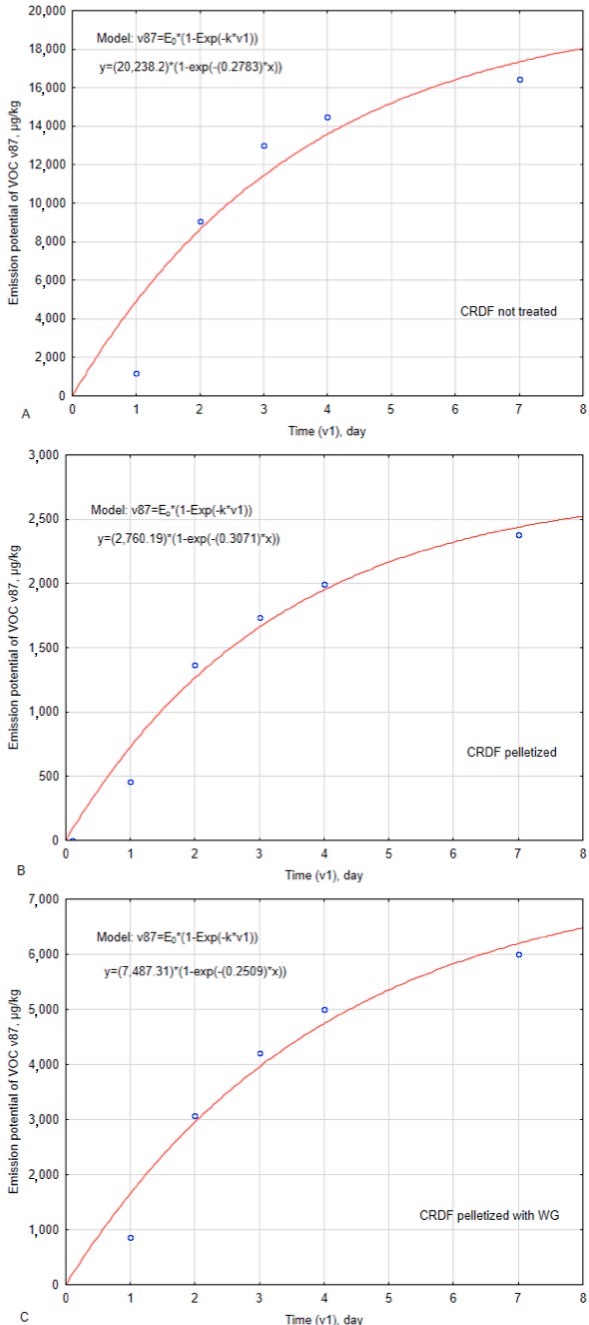

**Figure 1.** First-order kinetic emissions of volatile organic compounds (VOCs) from carbonized refuse-derived fuel (CRDF) post-treated in different scenarios (**A**) CRDF not treated (control); (**B**) pelletized CRDF; (**C**) pelletized CRDF with a binder (WG = water glass).

The Carbonized Refuse-Derived Fuel pelletization increased emission-rate constant $k$ from 0.280 $d^{-1}$ to 0.307 $day^{-1}$ (Table 1). The lowest value of $k$ (0.251 $day^{-1}$) was found in the case of CRDF pelletized with a binder (Table 1). However, the obtained $k$ values did not greatly differ, $\pm10\%$ in comparison to ground CRDF. The structural-modification effect on maximum emission potential $E_0$ was much greater. In both treatment cases, i.e., pelletized CRDF, and CRDF pelletized with a binder,

a significant reduction of 86% and 63% of VOC emissions in comparison to ground CRDF was observed, respectively (Table 1). This observation has great importance on the mitigation of VOCs emissions via pelletization for improving storage conditions and for reducing occupational exposure to VOC emissions from CRDF. Another useful observation pertains to half of the maximum potential of VOC emissions reaching between 2.26 and 2.76 days. Thus, it is recommended that the storage of sealed CRDF be relatively short due to the intensity of VOC emissions. Alternatively, a strategy to vent and treat VOC emissions from sealed storage should be developed. As structural modification mostly influenced the maximum emission potential, emission rate *r* was also reduced by 85% (CRDF pelletized), and 67% (CRDF pelletized with binder) in comparison to ground CRDF (Table 1).

**Table 1.** The volatile organic compounds emission kinetic parameters from ground (loose) CRDF (control), pelletized CRDF and pelletized CRDF with a binder (water glass, WG) ± standard error, with relative mitigation values, % obtained after structural modification in comparison to control CRDF (bold font).

| VOC Kinetic Parameters | Unit | CRDF Type | | |
|---|---|---|---|---|
| | | Ground (Control) | Pelletized (Treatment) | Pelletized with Binder (Treatment) |
| $E_0$ – maximum emission potential | $\mu g \cdot kg^{-1}$<br>% | $20{,}238 \pm 5848$<br>- | $2760 \pm 331$<br>**86.4** | $7487 \pm 1459$<br>**63.0** |
| $k$ – emission constant rate | $day^{-1}$<br>% | $0.280 \pm 0.152$<br>- | $0.307 \pm 0.073$<br>**−9.6** | $0.251 \pm 0.088$<br>**10.3** |
| $t_{0.5}$ – half time of emission | day<br>% | 2.5<br>- | 2.3<br>**8.8** | 2.8<br>**−11.6** |
| $r$ – emission rate | $\mu g \cdot (kg \cdot day)^{-1}$<br>% | 5666.6<br>- | 847.4<br>**85.0** | 1879.3<br>**66.8** |
| $R^2$ – determination coefficient | - | 0.876 | 0.976 | 0.947 |

VOC emission kinetic parameters from ground (loose) CRDF (control), pelletized CRDF and pelletized CRDF with a binder (WG) ± standard error, with relative mitigation values, % obtained after structural modification in comparison to control CRDF (bold font). Volatile organic compounds (VOCs), Carbonized refuse-derived fuel (CRDF).

## 4. Discussion

As many as 30 VOCs (out of 84 total) in all analyzed scenarios followed first-order kinetics for emissions in storage. Table A1 presents the summary of VOCs and their adherence to first-order kinetics. The apparent lack of adherence to the first-order kinetics model could be due to several reasons. First, some VOCs were measured at very low concentrations, and the apparent linear increase of cumulative emissions was observed (Supplementary Materials). The second reason pertains to VOCs that were initially emitted but, after the third or fourth day, a decrease in concentration was observed. SPME sampling could also be affected by competitive VOC adsorption and displacement from the SPME fiber coating [39–42]. This means that SPME sampling of VOCs could be affected by apparent higher emission rates and greater affinity of a particular VOC to the SPME fiber coating. This phenomenon should be further investigated. It was also found that some individual VOCs reacted differently to CRDF structural modification (data presented in Appendix A, Table A1).

The technology for real-time VOC detection and quantification in occupational settings is improving [43–45]. Sensors for the identification of explosion risks, including pellistors (thermal sensors) and nondispersive infrared (NDIR) sensors, are available and could be suitable for monitoring VOCs in storage. Metal oxide sensors (MOS), photoionization detectors (PIDs), and electrochemical sensors are suitable for VOC concentration range from ~1 ppmv to a low %, ~1 ppbv to below 1%, and low ppmv levels, respectively [43]. Conventional chromatographic analysis requires sophisticated equipment as well as surely being non-online. On the one hand, the constant reduction of gas chromatograph-mass spectrometer (GC–MS) prices and low-budget apparatus, available for less than €50,000, can be observed. Nowadays, there are also developed

modifications of chromatographical equipment (fast gas chromatography - FAST-GC or Fast Gas Chromatography-Mass Spectrometry—FAST-GC/MS) that allow complete analysis within a few minutes (e.g., a whole measurement time of around 7 min) [46]. Likewise, solid phase microextraction fast gas chromatography-mass spectrometry (SPME–FAST-GC/MS) systems were applied for trace aroma compound analysis [47]. On the other hand, fully automated systems for volatile analysis were demonstrated by Noventa et al. [48]. Nearly online (real-time) analyses could facilitated by MS systems [49], also equipped with a headspace module and separation capability. Miniaturization of single modules fabricated using microsystem technology, and even the first chip-scale mass spectrometers [50–52] is currently feasible. This could be the direction of detection method development, but it should be emphasized that the aforementioned systems are not yet commercially available.

In addition, we tested the hypothesis if increasing VOC boiling point is correlated with the decrease of the constant rate of emission. This was tested on VOCs for which emission kinetics followed the first-order model (i.e., $R^2 > 0.75$—supplementary materials, Table A1). In all cases, the determination coefficient was very low ($< 0.011$, Figure A1). This indicates that the measured emission rate is not solely related to VOC evaporation from CRDF surfaces. Other factors, such as high porosity, heterogeneity of the CRDF surface, the presence of numerous functional groups, and other physical and chemical interactions affecting the overall mass transfer of particular VOCs, should be further investigated.

Our experiment had a 'black-box' character showing the global effect of all phenomena, and was not designed to explain the reasons for the observed emission kinetics. Our experiment was rather exploratory in this field and aimed at testing practical approaches to VOC mitigation. We recommend continuing the studies on the mechanism of VOC emissions from CRDF, and other types of biochar, with consideration of the influence of structural and chemical biochar post-treatment. In our opinion, this could open a wide area for investigation both in fundamental science, e.g., explanation of the VOC emission mechanism, and applied science, e.g., scaling up the system for investigating the potential impact of VOCs on workers during CRDF storage, and methods of mitigation of VOC emissions as part of MSW torrefaction-technology development.

## 5. Conclusions

This research on VOC emission kinetics from stored CRDF in relation to its structural modification indicated:

- A significant effect of CRDF densification was observed for the maximum emission of total VOC potential, where pelletization decreases maximum emission potential $E_0$ by 86%, while pelletization with a binder reduced $E_0$ by 63%;
- pelletization both with and without a binder modified total VOC emission constant rate $k$ by only $\pm 10\%$ in relation to ground CRDF;
- half of maximum VOC potential was released within 2.26 to 2.76 d of storage. Therefore, it is recommended that shorter storage, and potential for venting and treating VOC emissions from CRDF should be explored;
- numerous deviations of emission patterns from the first-order model were noted for individual VOCs. More research in this area is warranted;
- a correlation between VOC boiling point and emission constant rate was not confirmed in all structural CRDF modification cases. More research in this area is warranted; and
- further research on the VOC emission mechanism from CRDF and other biochar types should be developed as a new niche in fundamental and applied biomass science, and waste conversion into high-quality solid fuels with consideration of worker-safety aspects.

**Supplementary Materials:** The following are available online at http://www.mdpi.com/2071-1050/11/3/935/s1. The following files were submitted as supplementary materials in zipped folder "supplementary materials.zip":

CRDF VOCs emission kinetics Sustainability.xlsx, containing detailed results of VOC emissions from stored CRDF, and the emission kinetic modeling.

**Author Contributions:** Conceptualization, A.B. and A.S.; methodology, A.B., M.M., A.S., and J.L; formal analysis, A.B., M.M., and J.L.; validation, A.B., M.M., A.S., and J.K.; investigation, M.M. and J.L.; resources, A.B. and A.S.; data curation, M.M., J.L., A.S., and A.B.; writing—original draft preparation, A.B. and M.M.; writing—review and editing, A.B., J.K., A.S., and J.L.; visualization, A.B. and J.K.; and supervision, A.B., A.S., and J.K.

**Funding:** Authors would like to thank the Fulbright Foundation for funding the project titled "Research on pollutant emissions from Carbonized Refuse Derived Fuel into the environment", completed at Iowa State University. In addition, this project was partially supported by the Iowa Agriculture and Home Economics Experiment Station, Ames, Iowa. Project no. IOW05556 (Future Challenges in Animal Production Systems: Seeking Solutions through Focused Facilitation) was sponsored by the Hatch Act and State of Iowa funds. The publication is financed under the program of the Minister of Science and Higher Education "Strategy of Excellence - University of Research" in 2018 - 2019 project number 0019 / SDU / 2018/18 in the amount of PLN 700 000.

**Conflicts of Interest:** The authors declare no conflict of interest. The funders had no role in the design of the study; in the collection, analyses, or interpretation of data; in the writing of the manuscript; or in the decision to publish the results.

## Appendix A The Deviations of VOCs Emissions Course from the First Order Reaction

As it was assumed that, when $R^2$ is higher than 0.75, the first-order emission model is accepted, individual compound emissions had a different character, including the influence of structural modification. In the case of all types of tested CRDF, the emission of the following compounds had an I-order character (Table A1): propanoic acid, pyrimidine, toluene, hexanal, 2-methylpyrazine, furan-2-carbaldehyde, 1,3-xylene, 2-oxopropyl acetate, 1,4-xylene, unknown compound, heptanal, 1-(furan-2-yl)ethenone, 4,6,6-trimethylbicyclo[3.1.1]hept-3-ene, n-propylbenzene, benzaldehyde, 1,3,5-trimethylbenzene, phenol, an unknown isomer of ethyl-dimethyl benzene, 2-ethyl-1,4-dimethylbenzene, 1-ethenyl-2,4-dimethylbenzene, 2-ethyl-1,4-dimethylbenzene, undecane, nonanal, 1,2,3,5-tetramethylbenzene, 1,3-dimethyl-2,3-dihydro-1H-indene, 1,3-diethyl-5-methylbenzene, 1-methyl-1H-indene, decanal, hexylbenzene, and 6-methyl-1,2,3,4-tetrahydronaphthalene.

It was also found that emission from all CRDF type variants of the following compounds differed from the first-order kinetics model (Table A1): 2-methylpropanoic acid, pentanoic acid, 1,2-xylene, 2-ethylpyrazine, 3-methylbutanoic acid, 1,3-diethylbenzene, 1-methyl-4-propan-2-ylbenzene, an unknown isomer of diethyl methylbenzene, 1,4-diethyl-2-methylbenzene, 5-methyl-1,2,3,4-tetrahydronaphthalene, 2-methyl-5-propan-2-ylphenol, 1-methylnaphtalene, and 3,3-dimethyl-2H-inden-1-one.

We found that:

- For ground CRDF and pelletized CRDF, the first-order emission model was not found for the following compounds (Table A1): pyridine, hexa-2,4-diene, (E,E)-, cumene, azulene, 4-methyl-2,3-dihydro-1H-indene, 1-methyl-4-propan-2-yl-2-[(E)-prop-1-enyl]benzene, and 5,6-dimethyl-1,2,3,4-tetrahydronaphthalene.

- For pelletized CRDF and pelletized CRDF with a binder, the first-order emission model was not found for the following compounds (Table A1): 2 and 3 case: 5-methylfuran-2-carbaldehyde, 4-methyl-1-propan-2-ylcyclohexene, 1,2-diethylbenzene, 1-methyl-2-propylbenzene, 2,4-diethyl-1-methyl benzene, and unknown compound.

- For ground CRDF and pelletized CRDF with binder the 1st order emission model was not found for following compounds (Table A1): dec-3-yn-1-ol, 2,3-dihydro-1H-indene, 1-phenylethanone, 2-ethyl-1,3-dimethylbenzene.

- For ground CRDF the first-order emission model was not found for the following compounds (Table A1): 2,5-dimethylpyrazine, 1,4-dimetylopirydyne, 4-ethyl-1,2-dimethylbenzene, 2-methoxyphenol, 1-undecyne, methyl benzoate, 5-methyl-2,3-dihydro-1H-indene, and 1,5-dimethyl-1,2,3,4-tetrahydronaphthalene.

- For pelletized CRDF, the first-order emission model was not found for the following compounds (Table A1): heptan-2-one, styrene, 4-ethylpyridine, 1,2,4,5-tetramethylbenzene, unknown compound,

pentylbenzene, 1,2,3,4-tetrahydronaphthalene, 2-ethyl-2,3-dihydro-1H-indene, and 4,7-dimethyl-2,3-dihydro-1H-indene.

- For pelletized CRDF with a binder, an I-order emission path was not found for the following compounds (Table A1): acetic acid, pentan-1-ol, 1,2,4-trimethylbenzene, octanal, 1-methyl-4-prop-1-en-2-ylcyclohexene, butylbenzene, and 1-ethyl-3,5-dimethylbenzene.

**Table A1.** The deviations of the course of the emission from the reaction of the first order.

| Compound Name (IUPAC) | Ground CRDF | Pelletized CRDF | Pelletized with WG CRDF |
|---|---|---|---|
| acetic acid | | | - |
| propanoic acid | | | |
| pyrimidine | | | |
| pyridyne | - | - | |
| pentan-1-ol | | | - |
| toluene | | | |
| 2-methylpropanoic acid | - | - | - |
| hexanal | | | |
| 2-methylpyrazine | | | |
| furan-2-carbaldehyde | | | |
| 1,3-xylene | | | |
| 2-oxopropyl acetate | | | |
| 1,4-xylene | | | |
| pentanoic acid | - | - | - |
| unknown compound | | | |
| heptan-2-one | | - | |
| styrene | | - | |
| 1,2-xylene | - | - | - |
| heptanal | | | |
| hexa-2,4-diene, (E,E)- | - | - | |
| 1-(furan-2-yl) ethanone | | | |
| 2-ethylpyrazine | - | - | - |
| 2,5-dimethylpyrazine | - | | |
| cumene | - | - | |
| 1,4-dimetylopirydyne | - | | |
| 4,6,6-trimethylbicyclo[3.1.1]hept-3-ene | | | |
| 3-methylbutanoic acid | - | - | - |
| 4-ethylpyridine | | - | |
| n-propylbenzene | | | |
| benzaldehyde | | | |
| 5-methylfuran-2-carbaldehyde | | - | - |
| 1,3,5-trimethylbenzene | | | |
| phenol | | | |
| 4-methyl-1-propan-2-ylcyclohexene | | - | - |
| 1,2,4-trimethylbenzene | | | - |
| octanal | | | - |
| dec-3-yn-1-ol | - | | - |
| an unknown isomer of ethyl-dimethyl benzene | | | |
| 1,3-diethylbenzene | - | - | - |
| 1-methyl-4-propan-2-ylben | - | - | - |
| 1-methyl-4-prop-1-en-2-ylcyclohexene | | | - |
| 2,3-dihydro-1H-indene | - | | - |
| 1,2-diethylbenzene | | - | - |
| 1-methyl-2-propylbenzene | | - | - |
| butylbenzene | | | - |
| 1-ethyl-3,5-dimethylbenzene | | | - |
| 2-ethyl-1,4-dimethylbenzene | | | |
| 1-phenylethanone | - | | - |
| 2-ethyl-1,3-dimethylbenzen | - | | - |
| 4-ethyl-1,2-dimethylbenzen | - | | |
| 1-ethenyl-2,4-dimethylbenzene | | | |
| 2-ethyl-1,4-dimethylbenzene | | | |
| 2-methoxyphenol | - | | |
| 1-undecyne | - | | |
| methyl benzoate | - | | |
| undecane | | | |
| nonanal | | | |
| 1,2,4,5-tetramethylbenzene | | - | |
| an unknown isomer of diethyl methylbenzene | - | - | - |
| unknown compound | | - | |
| 1,2,3,5-tetramethylbenzene | | | |
| 1,3-dimethyl-2,3-dihydro-1H-indene | | | |
| 5-methyl-2,3-dihydro-1H-in | - | | |
| 1,3-diethyl-5-methylbenzene | | | |
| 4-methyl-2,3-dihydro-1H-in | - | - | |

**Table A1.** *Cont.*

| Compound Name (IUPAC) | Ground CRDF | Pelletized CRDF | Pelletized with WG CRDF |
|---|---|---|---|
| 1-methyl-1H-indene | | | |
| pentylbenzene | | - | |
| 1,2,3,4-tetrahydronaphthalene | | - | |
| 1,4-diethyl-2-methylbenzen | - | - | - |
| 2,4-diethyl-1-methylbenzene | | - | - |
| azulene | - | - | |
| 1-methyl-4-propan-2-yl-2-[ benzene | - | - | |
| 2-ethyl-2,3-dihydro-1H-indene | | - | |
| decanal | | | |
| unknown compound | | - | - |
| hexylbenzene | | | |
| 6-methyl-1,2,3,4-tetrahydronaphthalene | | | |
| 5-methyl-1,2,3,4-tetrahydro | - | - | - |
| 4,7-dimethyl-2,3-dihydro-1H-indene | | - | |
| undecan-2-one (internal standard) | | | |
| 2-methyl-5-propan-2-ylphe | - | - | - |
| 1-methylnaphtalene | - | - | - |
| 3,3-dimethyl-2H-inden-1-o | - | - | - |
| 1,5-dimethyl-1,2,3,4-tetrahy | - | | |
| 5,6-dimethyl-1,2,3,4-tetrahy | - | - | |
| Total | | | |

Note: white, course in accordance with I-order; gray, deviation from the I-order path in all three variants of CRDF structural modification; blue, deviation from the I-order path in ground CRDF and pelletized CRDF variants; orange, deviation from the I-order path in pelletized CRDF and pelletized CRDF variants; yellow, deviation from the I-order path in ground CRDF and pelletized CRDF with water-glass bonder variants; green, deviation from the I-order path in ground CRDF variant; dark-blue, deviation from the I-order path in pelletized CRDF variant; and purple, deviation from the I-order path in pelletized CRDF with water-glass binder variant.

## Appendix B  The Visualization of the Correlation between VOC Boiling Points and Emission Constant Rates

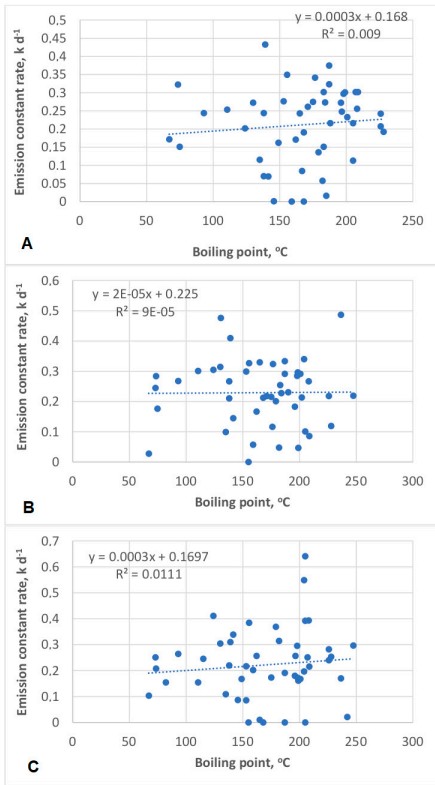

**Figure A1.** Correlation between VOC boiling point and emission constant rate k for each structural modification scenario: (**A**) ground CRDF—control; (**B**) pelletized CRDF, and (**C**) pelletized CRDF with WG.

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
