# Peer review of "Waste to Carbon: Influence of Structural Modification on VOC Emission Kinetics from Stored Carbonized Refuse-Derived Fuel"

_sustainability, doi:10.3390/su11030935_

Reviewer 1 Report

Dear Authors,   

    I find the results presented in the manuscript of interest for the Sustainability journal readers. The deep, but concise presentation of the data on CRDF (as resulted or treated) related with the VOC emission, from both literature and current research, might have potential use in industry. To strength the information presented in the manuscript, please find my minor comment below:

Line 58-59- I suggest considering also the following studies that support your statement: Rada, E. C.; Ragazzi, M.; Torretta, V.; Castagna, G.;Adami, L.; Cioca, L.I.; Circular economy and waste to energy, AIP Conference Proceedings 1968, 030050 (2018); https://doi.org/10.1063/1.5039237Rada, E. C.; Ionescu, G.;  Conti, F.; Cioca, L. I.;Torretta, V. (2018). Energy from Municipal Solid Waste: Some Considerations on Emissions and Health Impact. Quality-Access to Success, 19(167), http://www.srac.ro/calitatea/en/arhiva/2018/2018-06-Abstracts.pdf. Ionescu, G.; Rada, E. C.; Ragazzi, M.; Mărculescu, C.; Badea, A.; Apostol, T. (2013). Integrated municipal solid waste scenario model using advanced pretreatment and waste to energy processes. Energy Conversion and Management, 76, 1083-1092. doi: 10.1016/j.enconman.2013.08.049.

Best regards,

Author Response

Our replay is in the attached file.

Reviewer 2 Report

The presented work for the review is interesting and quite original, however, the authors themselves have pointed out that additional tests should be carried out to confirm the various hypotheses.

I believe that as a work presenting a problem and an environmental aspect, it is worth qualifying for further stages of evaluation. Below are my other remarks for this work:

line 71, I suggest inserting a table with heating values for individual waste that can be converted into fuel so that you can see the benefits of creating CRDF.

line 96-116, It is also worth mentioning in this paragraph at which concentration levels there were compounds from the VOC group and whether the admissible values were exceeded. In addition, there is no information what techniques have been used to identify and identify VOCs. Please correct this paragraph.

line 118-135, It would be worth placing a table with the advantages and disadvantages of CRDF pelletization in this paragraph

line 152-159, Extend this paragraph and describe the measurement methodology.

line 181-182, What is the standard deviation of the obtained constant emission rates ?? maybe statistically the differences are irrelevant?

Table 1, Please provide an estimation error for individual parameters.

line 218-221, Chromatographic tests are extremely expensive and time-consuming, while testing emission kinetics it would be worth using sensor systems, it would be possible to monitor on-line. It's worth emphasizing at work. I also suggest quoting works on sensor systems:

1. Application of electrochemical sensors and sensor matrixes for measurement of odorous chemical compounds, Trends in Analytical Chemistry, 77, 1-13, 2016

2. Review of Portable and Low-Cost Sensors for the Ambient Air Monitoring of Benzene and Other Volatile Organic Compounds, Sensors 2017, 17(7), 1520

3. Building energy metering and environmental monitoring – A state-of-the-art review and directions for future research, Energy and Buildings 2016, 120,85-102

4. Currently Commercially Available Chemical Sensors Employed for Detection of Volatile Organic Compounds in Outdoor and Indoor Air, Environments 2017, 4(1), 21

Author Response

Our replay is in the attached file.

Round  2

Reviewer 2 Report

All replies were comprehensive and satisfying. I hope that in the near future another publication will be created which will be an extension of the information contained in this publication, just as the authors declare here.

I recommend this manuscript for the next stages of evaluation.